# Green Credit Policy and Investment Decisions: Evidence from China

**Xiaoting Ling [1], Lijuan Yan [2],*** **and Deming Dai [1]**

1   Renmin Business School, Renmin University of China, Beijing 100872, China; lingxiaoting@ruc.edu.cn (X.L.); ddm_ruc@163.com (D.D.)
2   Management College, Beijing Union University, Beijing 100101, China
*   Correspondence: gltyanlijuan@buu.edu.cn

**Abstract:** Previous studies have reported mixed results on the effect of the green credit policy on firms' behaviors. Investment decision making is one of the most important elements of firms' behaviors, but few studies have discussed the relationship between the green credit policy and firms' investment decisions. Therefore, this paper explores the effect of green credit policy on firms' investment decisions. Using Chinese listed firms from 2008 to 2020, we found that the green credit policy tended to reduce pollutant-emitting firms' investment level but increases pollutant-emitting firms' investment efficiency; this effect was more pronounced in state-owned firms, firms with high-quality corporate governance, and those with a higher analyst following. This paper contributes to the literature on the economic consequences of the green credit policy and can help commercial banks and other financial institutions allocate green credits more effectively.

**Keywords:** green credit policy; investment level; investment efficiency; property rights; corporate governance; analyst following

## 1. Introduction

Recently, environmental problems, such as pollutant discharge and resource depletion, have become severe. Several countries have implemented policies to protect the environment. Environmental, social, and governance (ESG) activities have gradually attracted attention from the public and scholars. Existing research shows that firms have incentives to engage in ESG activities, as ESG activities can strengthen interactions with firms' stakeholders and enhance firms' reputation with the public [1]. Mason (2012) [2] found that when firms participated in ESG activities, customers would be willing to pay a higher premium on their products. Sassen et al. (2016) [3] indicated that firms with better ESG performance would have higher information transparency and a lower risk level. Fatemi et al. (2018) [4] suggested that ESG performance was positively related to firms' value. The stock market also values firms' ESG activities. Deng and Cheng (2019) [5] suggested that firms' ESG quality was positively related to their stock market performance, and the influence of the ESG quality on non-state-owned firms was greater than on state-owned firms.

As an important element of ESG practices, the green credit policy is a financial innovation to alleviate industrial pollution [6]. Specifically, the green credit policy requires commercial banks to provide preferential interest rates and sufficient credits for environmentally friendly firms and to restrict loans for heavily polluting firms [7,8]. Starting in 2007, China promulgated several green credit policies to optimize credit allocation and develop a more sustainable economy. Specifically, the Chinese government issued the 2012 Green Credit Guideline and 2016 Guidance on Building a Green Financial System. The 2012 Green Credit Guideline encouraged commercial banks to limit credits for pollutant-emitting firms. The 2016 Green Financial System required commercial banks to consider firms' environmental information when offering loans, and if commercial banks would

not consider the lending firms' environmental information when providing the credits, the commercial banks would be punished by the banking regulatory commission.

Although existing research shows that ESG activities are positively related to firms' performance [2,4,5], the green credit policy is an important part of ESG practice, but previous studies have not yielded conclusive results on the effect of the green credit policy on firms' behaviors. Some studies have suggested that firms benefit from the green credit policy, as they use the green credit to invest in environmental protection projects, which can provide stable profits in the long term and have a positive effect on the firms' performance [8,9]. Other studies found that the green credit policy negatively affected pollutant-emitting firms' behaviors. Zhang et al. (2011) [10] and Wang et al. (2020) [11] reported that the green credit policy restricted bank loans to pollutant-emitting firms, which led them into financial trouble. Wen et al. (2021) [12] found that the green credit policy had a negative effect on firms' total factor productivity.

This paper took Chinese green credit as the exogenous shock and investigated whether it could affect pollutant-emitting firms' investment decisions. There are two significant differences between this study and prior studies. On the one hand, the conflicting evidence of existing studies may be due to the fact that some research treated the voluntary green credit policy as the exogenous shock and investigated the causal effect of the green credit policy on firms' behaviors. Under the voluntary green credit policy, commercial banks would not always follow the green credit policy or would only limit credits on pollutant-emitting firms with poor performance; therefore, it is reasonable that the green credit policy might not affect firms' behaviors or might have a negative effect; this evidence cannot prove a causal link between the green credit policy and firms' behaviors. Different from existing studies, this paper took the compulsory green credit policy as an exogenous shock and investigated the causal effect of the green credit policy on firms' investment decisions. On the other hand, existing studies have discussed the effect of the green credit policy on firms' financial pressure, research and development expenditure, and total factor productivity [8,10,13,14], but few studies have discussed the effect of the green credit policy on firms' investment decisions. Firms' investment decisions are significantly affected by bank loans [15,16], and firms with limited credits from commercial banks would limit their investments [17], but it is unclear whether the firms would use the limited credits to make more efficient investment decisions.

Using Chinese listed firms from 2008 to 2020, the empirical results show that, after the 2016 Guidance on Building a Green Financial System was issued, pollutant-emitting firms received fewer bank loans from commercial banks and, thus, decreased their investment level but significantly increased their investment efficiency. These results were valid after we applied the propensity score matching (PSM) procedure and other robustness tests. The effect of the green credit policy on pollutant-emitting firms' investment decisions was more pronounced on state-owned firms, firms with high-quality corporate governance, and those with higher analyst following.

This paper makes several contributions, as follows. First, it investigated the effect of the Chinese green credit policy on firms' investment decisions, which enriches the literature on the economic consequences of ESG activities. Previous studies have shown that firms benefit from ESG activities, and the green credit policy is an important ESG activity, although existing research reveals mixed results on the effect of the green credit policy on firms' behaviors [7,8,11,12]. This paper showed that the compulsory green credit policy can reduce firms' investment level but increase their investment efficiency, which provides new evidence of the economic consequences of the green credit policy at the investment decision level. Second, the existing literature indicates that firms' investment level and investment efficiency are affected by different factors such as bank loans [16,17], property rights [10,18], and financial quality [19]. This paper showed that firms' investment level and investment efficiency were also affected by the green credit policy, which would complement the existing literature on investment decision making. Third, this paper addressed the implications of the practice. The empirical results suggest that the Chinese

green credit policy positively affects firms' investment efficiency. However, this positive effect was different for firms with varying property rights, corporate governance quality, and levels of analyst following. To improve green credit policy effectiveness, commercial banks can consider firms' characteristics when issuing green credits, and policy makers in other countries can consider changing the voluntary green credit policy to a compulsory policy.

## 2. Background, Literature Review, and Hypothesis Development

### 2.1. Background on Green Credit Policies

A green credit policy is a series of credit policies issued by financial institutions to promote energy conservation and emission reduction. Germany established the world's first environmental protection bank responsible for providing preferential loans for environmental projects in 1974 [20]. The Comprehensive Environmental Response issued by the USA, in 1980, stipulated that commercial banks need to pay attention to environmental pollution when issuing loans. Since then, the UK, Japan, and other countries have created green credit policies to encourage commercial banks to provide green credits to support green energy projects [21].

In 2007, China promulgated the green credit policy, which required commercial banks to consider lending firms' environmental protection track record. In 2012, the China Banking Regulation Commission (CBRC) released the Green Credit Guideline. The 2012 Green Credit Guideline encouraged financial institutions to issue green credits, and these green credits were provided to lending firms with better ESG performance and supported the development of a green and low-carbon economy. In 2016, the People's Bank of China (PBC), jointly with six other ministries and commissions, issued the Guidance on Building a Green Financial System, which urged commercial banks to boost green credits and curb pollutant-emitting industries' credits. Specifically, the 2016 Guidance on Building a Green Financial System stipulated that commercial banks should consider enterprises' environmental information, such as environmental violations, when granting credits, and the PBC would use the green credits to evaluate the commercial bank. The 2016 Guidance on Building a Green Financial System significantly changed commercial banks' credit allocation policy, making green credits a compulsory evaluation index for commercial banks. In 2018, the Tianjin Banking Regulatory Bureau imposed a fine of 500,000 yuan on Ping An Bank due to the fact of providing credits to enterprises that did not meet the environmental protection standards. Thus, after the 2016 Guidance on Building a Green Financial System, commercial banks have paid more attention to the green credit allocation and should significantly reduce pollutant-emitting industries' credit because of the PBC evaluation. In the rest of this paper, the 2016 Guidance on Building a Green Financial System is referred to as the 2016 Green credit policy.

### 2.2. Literature Review

As an important part of ESG practice, a green credit policy aims to protect the environment through reallocating bank loans among firms. Experts around the world have produced relevant research on the green credit policy. One strand of the literature has investigated whether commercial banks would follow the green credit policy. Generally, commercial banks are under greater scrutiny from regulators and the media as they dominate the allocation of credits in the economy, and they are attentive to the risk management and efficiency of credits allocation. Some studies have found that when commercial banks incorporate environmental sustainability into their lending policy, they are less exposed to information risks [22] and can develop a better reputation with the public [23,24]. Therefore, commercial banks are willing to follow the green credit policy, and their performance improved significantly after the implementation of the green credit policy [22,25]. Xing et al. (2021) [26] also found that Chinese commercial banks would consider firms' environmental information when issuing loans after the CBRC released the 2012 Green Credit Guideline. However, other studies suggested that the green projects invested by the green credits

cannot provide a higher return in the short term and, as a result, some commercial banks lack the economic incentive to comply with the green credit policy [13,14].

Another strand of the literature has investigated the effect of the green credit policy on firms' behaviors in nonfinancial industries. Some studies have suggested that the green credit policy has a positive effect on firms' behaviors. When firms with good ESG performance obtain sufficient green credits from commercial banks, they can use the green credits to invest more funds into environmental protection projects, and these green projects can provide stable profits in the long term and have a positive effect on firms' performance [22,27]. Cui et al. (2022) [8] found that heavily polluting firms benefited from the compulsory green credit policy. Specifically, after the green credit policy promulgation, the heavily polluting firms that received limited credits from commercial banks were forced to upgrade their facilities and improve production efficiency. However, another strand of the literature has indicated that the green credit policy has a negative effect on firms' behaviors. As the green credit policy restricts bank loans to firms with highly polluting projects, pollution-emitting firms have experienced higher financial pressure after the green credit policy promulgation [10,11]. Yao et al. (2021) [7] found that after the green credit policy was enacted, pollution-emitting firms without sufficient credits reduced their research and development intensity. Wen et al. (2021) [12] showed that the green credit policy significantly reduced pollutant-emitting firms' total factor productivity. In addition, some studies suggested that the green credit policy would not affect firms' behaviors, as the commercial banks would not follow the policy; thus, pollutant-emitting firms could still obtain sufficient credits and would not change their behaviors [14].

All told, the previous literature has not yielded consistent results on the economic consequences of the green credit policy. If commercial banks follow the green credit policy, pollutant-emitting firms should not obtain sufficient credits from commercial banks and will either use their limited loans to upgrade their facilities and participate in ESG practices or reduce their research and development investment. Thus, it is still unknown whether pollutant-emitting firms benefit from the green credit policy. Moreover, the existing literature has not discussed the effect of the green credit policy on firms' investment decisions. To fill this research gap, this paper took the 2016 Green credit policy as the exogenous shock and investigated the effect of Chinese green credit policy on firms' investment decisions.

*2.3. Hypothesis Development*

Previous studies suggest that if the green credit policy is a voluntary policy for commercial banks, banks will not always go along with it. Some studies found that commercial banks benefit from ESG practice [22,28] and have incentives to follow the green credit policy. Other research has suggested that commercial banks might not implement the green credit policy, as the ESG practice is a burden for the bank's operations [14].

The 2016 Green credit policy is a compulsory policy, requiring Chinese commercial banks to provide sufficient credits for green industries and to limit credits to pollutant-emitting firms. In addition, the 2016 Green credit policy took the green credit as the PBC's evaluation of commercial banks' performance. If commercial banks in China provide loans to firms that damage the environment, they will be punished by the PBC. Therefore, after the 2016 Green credit policy was implemented, Chinese commercial banks reduced credits to firms with a history of pollutant emissions.

Firms' investment level is affected by their credits from commercial banks [15]. Lemmon and Roberts (2010) [16] found that firms would inhibit their investment if they obtained fewer bank loans. Duchin et al. (2010) [17] indicated that when firms have limited external finances, they will reduce their investments significantly, and the relationship between the external finance and the investment level is more pronounced for firms with low cash reserves or high net short-term debt. After the 2016 Green credit policy, firms with a history of pollutant emissions had fewer credits from commercial banks, and a reduction in bank loans will lead to a reduction in pollutant-emitting firms' investments. Therefore, we propose the first hypothesis as follows:

**Hypothesis 1.** *The 2016 Green credit policy reduced pollutant-emitting firms' investment.*

It is believed that there is an optimal level of investment in firms that must be maintained to ensure investment efficiency [18]. However, firms will deviate their optimal investment level and will often be in a situation of overinvestment or underinvestment. Specifically, the agency problem shows that it is difficult for shareholders to supervise management, and management may invest in projects that are beneficial from management's perspective but detrimental from the perspective of shareholders [29]. For firms with abundant cash flow, management tends to overinvest in empire building. In contrast, the management of the firms with high leverage tends to give up valuable investment opportunities, since these projects provide fewer benefits to shareholders than to debtholders [29,30].

The 2016 Green credit policy led to decreased credit to pollutant-emitting firms, which would significantly affect the firms' investment efficiency. On the one hand, after the 2016 Green credit policy was released, pollutant-emitting firms had fewer credits from commercial banks, and management would be cautious about their investments and avoid blind expansion, decreasing their overinvestment. On the other hand, the 2016 Green credit policy drew more attention to firms' environmental risks; therefore, if pollutant-emitting firms wanted to obtain sufficient credits from commercial banks, they would need to use their limited cash flow to make valuable investments to develop new technology and increase the productivity of their regular projects [8]. Therefore, pollutant-emitting firms received fewer credits from commercial banks after the issuing of the 2016 Green credit policy. They were pushed to develop new technology, upgrade their technical facilities, and increase corporate productivity. As a result, pollutant-emitting firms needed to use their limited credits to make more efficient investments. We propose the second hypothesis as follows:

**Hypothesis 2.** *The 2016 Green credit policy increased pollutant-emitting firms' investment efficiency.*

### 3. Research Design

*3.1. Sample Selection and Data Source*

We analyzed data on firms publicly listed on the Shanghai Stock Exchange and Shenzhen Stock Exchange from 2008 to 2020. The initial sample comprised 27,485 observations. After excluding 739 sample observations from the financial industry according to the industry classification of the CSRC in 2012 and 1795 missing values, a final total of 24,951 sample observations were considered. The financial data were from the China Stock Market and the Accounting Research database (CSMAR), and firms' pollutant data were from Chinese Research Data Services (CNRDS). We Winsorized the sample data by 1% to eliminate the influence of outliers.

*3.2. Variable Definition*

3.2.1. Investment Level ($Inv_{i,t+1}$)

Following the prior literature [18,31], we estimated the investment level ($Inv_{i,t+1}$) as the cash payments for intangible assets, fixed assets, and other long-term assets minus the cash receipts from the sale of intangible assets, fixed assets, and other long-term assets, scaled by the total assets. The higher the $Inv_{i,t+1}$, the greater the investment level.

3.2.2. Investment Efficiency ($Inveff_{i,t+1}$)

We followed Zhang et al. (2019) [32] in measuring investment efficiency ($Inveff_{i,t+1}$) by the absolute value of the residual from the investment measurement model, as in Richardson (2006) [33], multiplied by −1. The greater the value of $Inveff_{i,t+1}$, the higher the investment efficiency.

### 3.2.3. Pollutant-Emitting Firms (Treat$_{i,t}$)

The 2016 Green credit policy required banks to provide green credits to the green industries and limit credits to pollutant-emitting firms. If firms emit pollutants, then firms are the treatment group that cannot obtain sufficient credits from the bank. Treat$_{i,t}$ is a dummy variable. Treat$_{i,t}$ equals 1 for firms with pollutant emissions; otherwise, it equals 0.

### 3.2.4. Event (Event$_{i,t}$)

The 2016 Green credit policy was considered as the exogenous shock. We used the dummy variable, Event$_{i,t}$, to represent the shock from the 2016 Green credit policy. Event$_{i,t}$ is an indicator variable that equals 1 for each year after the 2016 Green credit policy implementation and 0 otherwise.

### 3.2.5. Control Variables

Following the prior literature [11,27,34], we chose as controls the firm assets (Size), the ratio of net income to total assets (Roa), the debt-to-asset ratio (Lev), the cash holdings (Cash), the book-to-market ratio (Mb), the research and development expenditure (Rd), the stock returns (Yretwd), the board size (Board), the CEO duality (Dual), and the board independence (Indp). Detailed definitions of these variables are presented in Table 1.

**Table 1.** Variable definition.

| Variable | Definition |
|---|---|
| **Dependent variables** | |
| Inv | Total investments scaled by the total asset, estimated as the cash payments for intangible assets, fixed assets, and other long-term assets minus the cash receipts from the sale of intangible assets, fixed assets, and other long-term assets, scaled by the total assets. |
| Inveff | Using the absolute value of the residual from the Richardson (2006) [33] investment measurement model multiplied by $-1$ $$Inv_{i,t+1} = \alpha_0 + \alpha_1 Inv_{i,t} + \alpha_2 Size_{i,t} + \alpha_3 Leverage_{i,t} + \alpha_4 Growth_{i,t} + \alpha_5 Yrerwd_{i,t} + \alpha_6 Acash_{i,t} + \alpha_7 Age_{i,t} + \sum Industry_i + \sum Year_t + \varepsilon$$ |
| **Independent variables** | |
| Treat | An indicator variable that equals 1 if firm i emitted pollutants before the 2016 Green credit policy and 0 otherwise. |
| Event | An indicator variable that equals 1 for each year after the 2016 Green credit policy implementation and 0 otherwise. |
| **Control variables** | |
| Size | The natural logarithm of the total assets. |
| Roa | The ratio of net profit to the book value of total assets. |
| Lev | The ratio of total debt to total assets. |
| Cash | The natural logarithm of net cash flow. |
| Mb | The ratio of the book value of equity to the market value of equity. |
| Rd | The ratio of research and development expenditure to the total assets. |
| Yretwd | The stock returns of the year. |
| Board | The number of directors on the enterprise's board. |
| Dual | An indicator variable that equals 1 if the CEO serves as the chairman and 0 otherwise. |
| Indp | The ratio of the number of the enterprise's independent directors to the board directors. |

**Table 1.** *Cont.*

| Variable | Definition |
|---|---|
| Other variables | |
| Soe | An indicator variable that equals 1 if the enterprise's major shareholders are government entities and 0 otherwise. |
| Cgi | The corporate governance level is calculated as the sum of 14 internal governance factors as shown in Appendix A. |
| Ana | The ratio of the number of analysts' reports to the number of analysts. |
| Loginv | The natural logarithm of total investments, that is, the natural logarithm of the cash payments for intangible assets, fixed assets, and other long-term assets minus the cash receipts from the sale of intangible assets, fixed assets, and other long-term assets, scaled by the total assets. |
| Inveff1 | Using the absolute value of the residual from the investment measurement model as in Biddle et al. (2009) [19], multiplied by −1. |
| Acash | The ratio of the net cash flow at the beginning of the year to the total assets at the end of the year. |
| Age | The number of years from the IPO year to the financial reporting year. |

*3.3. Research Model*

To investigate the effect of the 2016 Green credit policy on investment decisions, we used a difference-in-difference (DID) methodology. Equations (1) and (2) examine whether the 2016 Green credit policy affected pollutant-emitting firms' investment level and investment efficiency.

In Equation (1), the dependent variable represents the investment level ($Inv_{i,t+1}$). The higher the $Inv_{i,t+1}$, the greater the investment level. In Equation (2), the dependent variable represents the investment efficiency ($Inveff_{i,t+1}$). The higher the $Inveff_{i,t+1}$, the greater the investment efficiency.

$$Inv_{i,t+1} = \beta_0 + \beta_1 Treat_{i,t} + \beta_2 Treat_{i,t} \times Event_{i,t} + \beta_3 Event_{i,t} + \beta_4 Controls_{i,t} + \sum Industry_i + \varepsilon \tag{1}$$

$$Inveff_{i,t+1} = \beta_0 + \beta_1 Treat_{i,t} + \beta_2 Treat_{i,t} \times Event_{i,t} + \beta_3 Event_{i,t} + \beta_4 Controls_{i,t} + \sum Industry_i + \varepsilon \tag{2}$$

The interaction term ($Treat_{i,t} \times Event_{i,t}$) captures the difference-in-difference (DID) effect in Equations (1) and (2). The DID effect is the differential change in the investment level and investment efficiency between pollutant-emitting and non-pollutant-emitting enterprises across the pre- and post-2016 Green credit policy periods. The 2016 Green credit policy limited pollutant-emitting firms' access to credit; without abundant bank loans, pollutant-emitting firms would be forced to decrease their investment level but increase their investment efficiency. Therefore, we expected the coefficient of $\beta_2$ in Equation (1) to be significantly negative, and the coefficient of $\beta_2$ in Equation (2) to be significantly positive.

In addition, $Controls_{i,t}$ in Equations (1) and (2) are control variables. To control the industry effect, following previous studies [9,35,36], Equations (1) and (2) take $\sum Industry_i$ to represent the industry dummy variable, which can control the events that may affect specific industries.

## 4. Empirical Results

*4.1. Descriptive Statistics*

Table 2 reports descriptive statistics for the main variables. The mean values for the investment level ($Inv_{i,t+1}$) and the investment efficiency ($Inveff_{i,t+1}$) were 0.05 and −0.03, respectively. The mean values for pollutant-emitting firms ($Treat_{i,t}$) were 0.09, indicating that 9% of the observations were pollutant-emitting firms that were affected by the 2016 Green credit policy. The mean values for firm size ($Size_{i,t}$), leverage level ($Lev_{i,t}$), and

cashflow ($Cash_{i,t}$) were 22.19, 0.45, and 19.94, respectively, and the average return on asset ($Roa_{i,t}$) was 0.05, indicating that the listed firms in our sample were large firms with relatively good performance and sufficient cash flow. The minimum board size ($Board_{i,t}$) was 4, the maximum ($Board_{i,t}$) was 18, and the mean values of the percentage of independent directors in the board ($Indep_{i,t}$) was 0.39. The descriptive statistics of other variables were similar to the previous literature [8,32].

**Table 2.** Descriptive statistics.

| Variable | $N$ | Mean | Median | SD | Minimum | Maximum |
|---|---|---|---|---|---|---|
| $Inv_{i,t+1}$ | 24,951 | 0.05 | 0.03 | 0.05 | 0.00 | 0.64 |
| $Inveff_{i,t+1}$ | 24,951 | −0.03 | −0.02 | 0.03 | −0.17 | 0.00 |
| $Treat_{i,t}$ | 24,951 | 0.09 | 0.00 | 0.28 | 0.00 | 1.00 |
| $Event_{i,t}$ | 24,951 | 0.44 | 0.00 | 0.50 | 0.00 | 1.00 |
| $Size_{i,t}$ | 24,951 | 22.19 | 22.03 | 1.29 | 19.57 | 26.10 |
| $Lev_{i,t}$ | 24,951 | 0.45 | 0.45 | 0.21 | 0.06 | 0.90 |
| $Roa_{i,t}$ | 24,946 | 0.05 | 0.06 | 0.17 | −1.08 | 0.33 |
| $Mb_{i,t}$ | 24,951 | 3.64 | 2.65 | 3.42 | 0.62 | 23.94 |
| $Cash_{i,t}$ | 24,951 | 19.94 | 19.88 | 1.38 | 16.39 | 23.79 |
| $Yretwd_{i,t}$ | 24,951 | 0.15 | 0.00 | 0.60 | −0.71 | 2.49 |
| $Rd_{i,t}$ | 24,951 | 0.01 | 0.00 | 0.01 | 0.00 | 0.70 |
| $Dual_{i,t}$ | 24,951 | 0.01 | 0.00 | 0.11 | 0.00 | 1.00 |
| $Board_{i,t}$ | 24,951 | 9.66 | 9.00 | 2.77 | 4.00 | 18.00 |
| $Indp_{i,t}$ | 24,951 | 0.39 | 0.38 | 0.99 | 0.00 | 0.67 |

Figures 1 and 2 show the investment level and investment efficiency of pollutant-emitting firms and non-pollutant-emitting firms from 2009 to 2020. In Figure 1, the *y*-axis is the investment level ($Inv_{i,t+1}$), which is scaled by the cash payments for fixed assets, intangible assets, and other long-term assets minus the cash receipts from the sale of intangible assets, fixed assets, and other long-term assets, divided by the total assets at the end of year. Figure 1 suggests that the investment level of the pollutant-emitting firms was higher than that of the non-pollutant-emitting firms before the 2016 Green credit policy, and the investment level of pollutant-emitting firms decreased significantly after 2016. In Figure 2, the *y*-axis is the investment efficiency ($Inveff_{i,t+1}$), which is the absolute value of the residual from the Richardson (2006) [33] model multiplied by −1. Figure 2 shows that the investment efficiency of the pollutant-emitting firms was lower than that of the non-pollutant-emitting firms before the 2016 Green credit policy, but the investment efficiency of the pollutant-emitting firms increased significantly and even exceeded that of non-pollutant-emitting firms after 2016. Figures 1 and 2 indicate that the 2016 Green credit policy can decrease pollutant-emitting firms' investment level but improve their investment efficiency, which provides preliminary evidence for confirming Hypotheses 1 and 2.

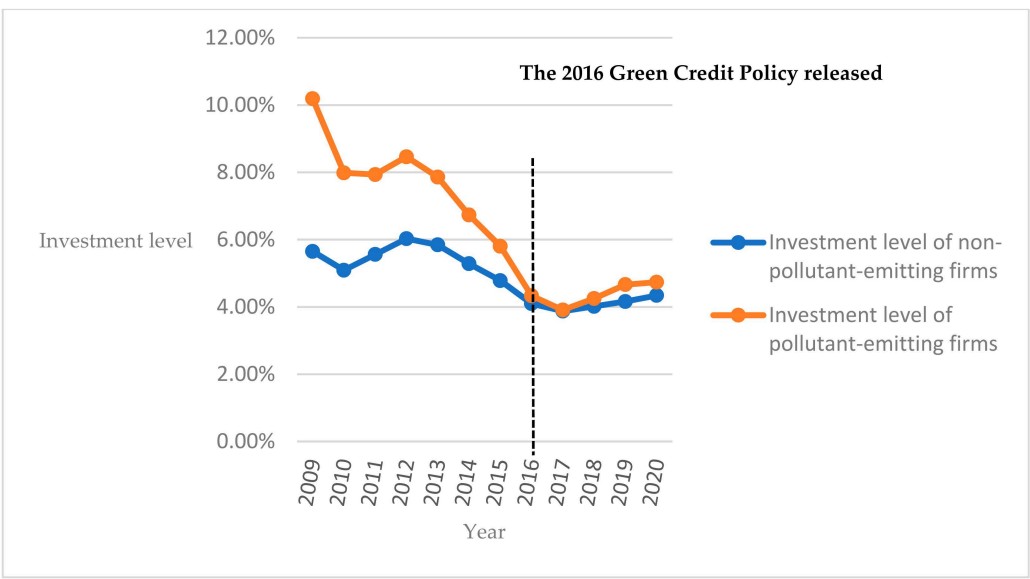

**Figure 1.** Investment level of pollutant-emitting firms and non-pollutant-emitting firms from 2009 to 2020. Note: The black dot line means the year 2016 Green Credit Policy was released.

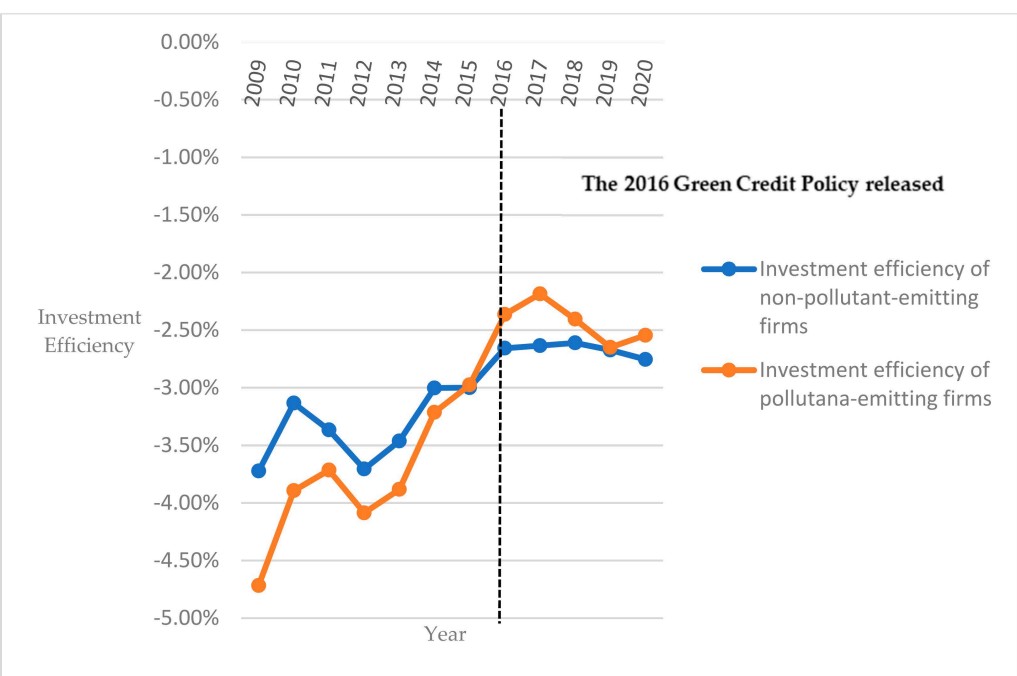

**Figure 2.** Investment efficiency of pollutant-emitting firms and non-pollutant-emitting firms from 2009 to 2020. Note: The black dot line means the year 2016 Green Credit Policy was released.

To ensure that Equations (1) and (2) have no multicollinearity problems, we provide the results of the Pearson and Spearman coefficients and the variance inflation factors. In Appendix A, the Pearson and Spearman coefficients between the variables in Equations (1) and (2) were all below 0.7. In Appendix B, the mean value of the variance inflation factors for Equations (1) and (2) were 1.91 and 1.92, respectively, and the variance inflation factors for the single variables were all below 5. Following Gujarati (2003) [37], the results in Appendices A and B suggest that there was no collinearity problem in Equations (1) and (2).

### 4.2. Empirical Regressions

Table 3 provides the regression results of Equations (1) and (2). In Table 3, column (1) provides the empirical results of Equation (1). The coefficients of $Treat_{i,t}$ and $Treat_{i,t} \times Event_{i,t}$ were 0.016 and −0.017, respectively, and both were significant at the 1% level, suggesting that after the implementation of the 2016 Green credit policy, pollutant-emitting firms received fewer bank loans and a higher interest rate and, thus, decreased their investment level. Therefore, Hypothesis 1 is supported.

**Table 3.** Green credit policy and investment decisions.

|  | (1) | (2) |
|---|---|---|
|  | $Inv_{i,t+1}$ | $Inveff_{i,t+1}$ |
| $Treat_{i,t}$ | 0.016 *** | −0.003 ** |
|  | (5.463) | (−2.112) |
| $Treat_{i,t} \times Event_{i,t}$ | −0.017 *** | 0.005 *** |
|  | (−5.715) | (2.863) |
| $Event_{i,t}$ | −0.013 *** | 0.006 *** |
|  | (−13.144) | (9.761) |
| $Size_{i,t}$ | −0.001 | −0.000 |
|  | (−1.442) | (−0.676) |
| $Lev_{i,t}$ | 0.006 ** | −0.005 *** |
|  | (1.972) | (−2.754) |
| $Roa_{i,t}$ | 0.030 *** | −0.009 *** |
|  | (13.628) | (−6.068) |
| $Mb_{i,t}$ | −0.000 | −0.000 *** |
|  | (−0.596) | (−4.511) |
| $Cash_{i,t}$ | 0.002 *** | 0.001 ** |
|  | (2.643) | (2.509) |
| $Yretwd_{i,t}$ | −0.003 *** | 0.003 *** |
|  | (−6.083) | (7.893) |
| $Rd_{i,t}$ | 0.108 *** | 0.006 |
|  | (3.783) | (0.399) |
| $Dual_{i,t}$ | 0.002 | −0.004 * |
|  | (0.478) | (−1.720) |
| $Board_{i,t}$ | −0.000 ** | 0.000 |
|  | (−2.018) | (1.500) |
| $Indp_{i,t}$ | −0.001 | −0.000 |
|  | (−0.163) | (−0.082) |
| $Constant_{i,t}$ | 0.059 *** | −0.045 *** |
|  | (4.981) | (−6.352) |
| Industry-fixed effects | Yes | Yes |
| Observations | 24,946 | 24,946 |
| Adjusted $R^2$ | 0.1050 | 0.0373 |

All models in Table 3 were estimated with industry-fixed effects. Reported in parentheses are the *t*-statistics based on robust standard errors clustered at the firm level. *, **, and *** indicate statistical significance at the 10%, 5%, and 1% levels, respectively.

In Table 3, column (2) gives the empirical results of Equation (2). In column (2), the coefficient of the interaction term ($Treat_{i,t} \times Event_{i,t}$) was 0.005 and statistically significant at the 1% level, while the coefficient of $Treat_{i,t}$ was −0.003 and statistically significant at the 5% level. The empirical results from column (2) indicate that after the commencement of the 2016 Green credit policy, pollutant-emitting firms received fewer bank loans and used the limited credits to increase their investment efficiency. Thus, Hypothesis 2 is supported.

### 4.3. Further Analysis

4.3.1. The Moderating Effects of Different Property Rights

This section examines whether the 2016 Green credit policy had different effects on Chinese state-owned enterprises (SOEs) and non-state-owned enterprises (non-SOEs). Pre-

vious studies found that SOEs can easily obtain substantial bank loans from state-owned banks, even if they exhibit poor performance [18]. In recent years, the Chinese government has paid more attention to firms' ESG performance and issued the green credit policy, which stipulated that commercial banks should consider firms' environmental performance when providing loans [10]. Yao et al. (2021) [7] and Zhang et al. (2022) [32] found that compared with non-SOEs, heavily polluting SOEs face severe economic penalties and public criticism if they do not undertake their social responsibilities. Therefore, after the commencement of the 2016 Green credit policy, considering the severe penalties, commercial banks provided fewer credits to pollutant-emitting SOEs compared with the pollutant-emitting non-SOEs, and pollutant-emitting SOEs have to make a greater effort to improve their social responsibilities and make more efficient investment decisions.

We divided our sample into an SOEs group and a non-SOEs group. We investigated the effects of the 2016 Green credit policy on state-owned enterprises and non-state-owned enterprises. Table 4 provides the empirical results. In column (1), the coefficients of $Treat_{i,t}$ and $Treat_{i,t} \times Event_{i,t}$ were 0.021 and $-0.022$; both were significant at the 1% level. The coefficient of the interaction term ($Treat_{i,t} \times Event_{i,t}$) in column (2) was negative and nonsignificant, while the coefficient of $Treat_{i,t}$ in column (2) was positive and statistically significant at the 5% level. The empirical results in columns (1) and (2) suggest that the investment level of SOEs decreased more significantly after the 2016 Green credit policy compared with the non-SOEs.

**Table 4.** The moderating effects of ownership.

| | (1) | (2) | (3) | (4) |
|---|---|---|---|---|
| | SOEs | Non-SOEs | SOEs | Non-SOEs |
| | $Soes_{i,t} = 1$ | $Soes_{i,t} = 0$ | $Soes_{i,t} = 1$ | $Soes_{i,t} = 0$ |
| | $Inv_{i,t+1}$ | $Inv_{i,t+1}$ | $Inveff_{i,t+1}$ | $Inveff_{i,t+1}$ |
| $Treat_{i,t}$ | 0.021 *** | 0.010 ** | −0.005 *** | −0.001 |
| | (5.814) | (2.178) | (−2.713) | (−0.526) |
| $Treat_{i,t} \times Event_{i,t}$ | −0.022 *** | −0.007 | 0.005 *** | 0.003 |
| | (−6.290) | (−1.348) | (2.725) | (0.822) |
| $Event_{i,t}$ | −0.016 *** | −0.014 *** | 0.007 *** | 0.006 *** |
| | (−10.441) | (−10.187) | (7.797) | (6.958) |
| $Size_{i,t}$ | −0.002 | 0.000 | −0.000 | −0.000 |
| | (−1.598) | (0.088) | (−0.559) | (−0.521) |
| $Lev_{i,t}$ | 0.015 *** | 0.006 | −0.006 ** | −0.006 *** |
| | (2.976) | (1.411) | (−2.092) | (−2.636) |
| $Roa_{i,t}$ | 0.030 *** | 0.027 *** | −0.012 *** | −0.006 *** |
| | (7.170) | (11.056) | (−5.195) | (−3.437) |
| $Mb_{i,t}$ | −0.000 | 0.000 | −0.000 ** | −0.001 *** |
| | (−1.419) | (0.289) | (−2.034) | (−3.825) |
| $Cash_{i,t}$ | 0.001 | 0.003 *** | 0.001 * | 0.000 |
| | (0.724) | (3.939) | (1.857) | (0.917) |
| $Yretwd_{i,t}$ | −0.005 *** | −0.003 *** | 0.003 *** | 0.003 *** |
| | (−6.093) | (−3.893) | (6.218) | (5.220) |
| $Rd_{i,t}$ | 0.042 | 0.103 *** | 0.047 | 0.007 |
| | (0.841) | (3.123) | (1.535) | (0.431) |
| $Dual_{i,t}$ | −0.003 | 0.001 | −0.006 | −0.001 |
| | (−0.361) | (0.210) | (−1.225) | (−0.649) |
| $Board_{i,t}$ | 0.000 ** | −0.001 *** | −0.000 | 0.000 |
| | (2.044) | (−3.066) | (−0.481) | (1.266) |
| $Indp_{i,t}$ | −0.011 | 0.002 | 0.002 | 0.001 |
| | (−1.581) | (0.315) | (0.406) | (0.192) |
| $Constant_{i,t}$ | 0.072 *** | 0.023 | −0.044 *** | −0.038 *** |
| | (4.819) | (1.221) | (−4.289) | (−3.757) |
| Industry-fixed effects | Yes | Yes | Yes | Yes |
| Observations | 10,769 | 14,177 | 10,769 | 14,177 |
| Adjusted $R^2$ | 0.1421 | 0.1012 | 0.0501 | 0.0335 |

All models in Table 4 are estimated with industry-fixed effects. Reported in parentheses are the *t*-statistics based on robust standard errors clustered at the firm level. *, **, and *** indicate statistical significance at the 10%, 5%, and 1% levels, respectively.

The coefficients of $Treat_{i,t}$ and the interaction term ($Treat_{i,t} \times Event_{i,t}$) in column (3) were −0.005 and 0.005, respectively, both significant at the 1% level. In column (4), the coefficients of $Treat_{i,t}$ and $Treat_{i,t} \times Event_{i,t}$ were nonsignificant. The empirical results in columns (3) and (4) show that the investment efficiency of SOEs increased more significantly after the 2016 Green credit policy compared with the non-SOEs.

4.3.2. The Moderating Effects of Corporate Governance

This section analyzes whether corporate governance played a moderating role on the effect of the 2016 Green credit policy on firms' investment decisions. Previous studies found that effective corporate governance can mitigate agency conflicts and curb value-destructive activities [38,39]. Specifically, high-quality governance may deter firms from pursuing poor investment decisions through imposing significant penalties on firms' ineffective investment decisions [40]. Therefore, after the 2016 Green credit policy, high-quality corporate governance would make pollutant-emitting firms use their limited credits more effectively and avoid suboptimal investment decisions.

Following Wu et al. (2020) [41] and Dai and Xue (2021) [36], we constructed the corporate governance level ($Cgi_{i,t}$) using 14 internal governance factors, including the strategy committee size, and nomination committee size, as shown in Appendix A. Specifically, we first estimated whether these 14 variables from firms were higher than the median value of the same industry in the same year. Secondly, if one of these 14 variables was higher than the median value of the same industry in the same year, then we gave one point to the firm; otherwise, we gave zero points. Taking the compensation committee size as an example, if the firm's compensation committee size was bigger than the median value in the same industry in the same year, then we gave this firm one point, while if the firm's compensation committee size was smaller than the median value in the same industry in the same year, we gave zero points. Finally, we added up the points that the firms obtained, and the total was the corporate governance level ($Cgi_{i,t}$). $Cgi_{i,t}$ represents firms' corporate governance quality: the higher the level of $Cgi_{i,t}$, the higher the level of corporate governance.

We used the median value of the $Cgi_{i,t}$ to divide the sample into high-quality and low-quality corporate governance. In Table 5, columns (1) and (2), the coefficients of $Treat_{i,t} \times Event_{i,t}$ were significant at the 1% and 5% levels, respectively. In Table 5, columns (3) and (4), the coefficients of $Treat_{i,t} \times Event_{i,t}$ were significant at the 1% level and nonsignificant, respectively. The results in Table 5 suggest that under different levels of corporate governance, the impact of the 2016 Green credit policy on the investment decisions of pollutant-emitting enterprises differed. Specifically, compared with pollutant-emitting firms with low-quality corporate governance, pollutant-emitting firms with high-quality corporate governance reduced their investment level and improved their investment efficiency more significantly after the 2016 Green credit policy.

**Table 5.** The moderating effects of corporate governance.

| | (1) | (2) | (3) | (4) |
|---|---|---|---|---|
| | High corporate governance quality | Low corporate governance quality | High corporate governance quality | Low corporate governance quality |
| | $Cgi_{i,t} >$ median value | $Cgi_{i,t} \leq$ median value | $Cgi_{i,t} >$ median value | $Cgi_{i,t} \leq$ median value |
| | $Inv_{i,t+1}$ | $Inv_{i,t+1}$ | $Inveff_{i,t+1}$ | $Inveff_{i,t+1}$ |
| $Treat_{i,t}$ | 0.017 *** | 0.012 *** | −0.004 ** | −0.001 |
| | (5.283) | (2.592) | (−2.271) | (−0.469) |
| $Treat_{i,t} \times Event_{i,t}$ | −0.020 *** | −0.010 ** | 0.006 *** | 0.003 |
| | (−5.800) | (−2.114) | (2.630) | (1.290) |
| $Event_{i,t}$ | −0.013 *** | −0.014 *** | 0.006 *** | 0.006 *** |
| | (−9.982) | (−9.484) | (7.477) | (6.635) |
| $Size_{i,t}$ | −0.001 | −0.001 | −0.001 | 0.000 |
| | (−0.593) | (−1.218) | (−1.489) | (0.534) |
| $Lev_{i,t}$ | 0.003 | 0.010 ** | −0.003 | −0.007 *** |
| | (0.772) | (2.477) | (−1.437) | (−2.837) |
| $Roa_{i,t}$ | 0.036 *** | 0.026 *** | −0.012 *** | −0.007 *** |
| | (9.629) | (10.304) | (−5.563) | (−3.568) |
| $Mb_{i,t}$ | 0.000 | −0.000 | −0.001 *** | −0.000 *** |
| | (0.140) | (−0.828) | (−3.397) | (−3.313) |
| $Cash_{i,t}$ | 0.000 | 0.003 *** | 0.001 *** | 0.000 |
| | (0.503) | (3.880) | (3.031) | (0.368) |
| $Yretwd_{i,t}$ | −0.004 *** | −0.003 *** | 0.003 *** | 0.003 *** |
| | (−5.202) | (−3.547) | (5.622) | (5.572) |
| $Rd_{i,t}$ | 0.103 ** | 0.103 *** | −0.003 | 0.012 |
| | (2.289) | (3.026) | (−0.132) | (0.591) |
| $Dual_{i,t}$ | 0.005 | −0.001 | −0.008 * | −0.001 |
| | (0.678) | (−0.259) | (−1.848) | (−0.371) |
| $Board_{i,t}$ | −0.000 | −0.000 ** | 0.000 | 0.000 |
| | (−0.803) | (−2.196) | (0.825) | (1.288) |
| $Indp_{i,t}$ | −0.001 | −0.001 | −0.001 | 0.000 |
| | (−0.137) | (−0.116) | (−0.145) | (0.044) |
| $Constant_{i,t}$ | 0.065 *** | 0.042 ** | −0.041 *** | −0.047 *** |
| | (4.398) | (2.492) | (−4.536) | (−4.974) |
| Industry-fixed effects | Yes | Yes | Yes | Yes |
| Observations | 13,231 | 11,715 | 13,231 | 11,715 |
| Adjusted $R^2$ | 0.1147 | 0.0965 | 0.0410 | 0.0341 |

All models in Table 5 were estimated with industry-fixed effects. Reported in parentheses are the *t*-statistics based on robust standard errors clustered at the firm level. *, **, and *** indicate statistical significance at the 10%, 5%, and 1% levels, respectively.

### 4.3.3. The Moderating Effects of Analyst Following

This section tests whether analyst played a moderating role on the effect of the 2016 Green credit policy on firms' investment decisions. The analyst plays a vital role in propagating bad news and mitigating information asymmetry [42]. Yao et al. (2021) [7] indicate that, as an external supervisory force, analyst following makes firms pay more attention to the long-term value brought about by environmental protection actions, and they found that the positive effect of the green credit policy on firms' performance was more significant in firms with a high analyst following. Thus, after the implementation of the 2016 Green credit policy, analyst following deterred pollutant-emitting firms from pursuing value-destroying activities and led to pollutant-emitting firms being more cautious regarding their investment decisions.

We measured the analyst following ($Ana_{i,t}$) by the ratio of the number of analysts' reports to the number of analysts according to Wu et al. (2020) [41]. Table 6 shows that the coefficients of the interaction term ($Treat_{i,t} \times Event_{i,t}$) in columns (1) and (2) were significantly negative at the 1% level. In columns (3) and (4), the coefficients of $Treat_{i,t} \times Event_{i,t}$ were significant at the 1% and 5% levels, respectively, indicating that compared with pollutant-emitting firms followed by fewer analysts, pollutant-emitting firms followed by more analysts increased their investment efficiency more significantly. The results in

Table 6 suggest that the analyst following cannot moderate the effect of the 2016 Green credit policy on the firms' investment level but can moderate the effect on the firms' investment efficiency.

**Table 6.** The moderating effects of analyst following.

| | (1) | (2) | (3) | (4) |
|---|---|---|---|---|
| | High analyst following | Low analyst following | High analyst following | Low analyst following |
| | $Ana_{i,t} >$ median value | $Ana_{i,t} \leq$ median value | $Ana_{i,t} >$ median value | $Ana_{i,t} \leq$ median value |
| | $Inv_{i,t+1}$ | $Inv_{i,t+1}$ | $Inveff_{i,t+1}$ | $Inveff_{i,t+1}$ |
| $Treat_{i,t}$ | 0.020 *** | 0.008 *** | −0.006 *** | 0.000 |
| | (5.654) | (2.758) | (−2.863) | (0.161) |
| $Treat_{i,t} \times Event_{i,t}$ | −0.018 *** | −0.014 *** | 0.005 *** | 0.005 ** |
| | (−4.748) | (−4.180) | (2.944) | (2.112) |
| $Event_{i,t}$ | −0.012 *** | −0.012 *** | 0.005 *** | 0.006 *** |
| | (−8.629) | (−9.704) | (5.803) | (7.686) |
| $Size_{i,t}$ | −0.003 ** | −0.003 *** | 0.000 | 0.000 |
| | (−2.571) | (−2.798) | (0.053) | (0.722) |
| $Lev_{i,t}$ | 0.011 ** | 0.011 *** | −0.009 *** | −0.003 |
| | (2.459) | (3.100) | (−3.451) | (−1.580) |
| $Roa_{i,t}$ | 0.037 *** | 0.017 *** | −0.016 *** | −0.004 ** |
| | (8.492) | (7.424) | (−5.803) | (−2.389) |
| $Mb_{i,t}$ | 0.000 | −0.001 *** | −0.000 ** | −0.000 *** |
| | (0.780) | (−2.692) | (−2.199) | (−3.305) |
| $Cash_{i,t}$ | −0.001 | 0.003 *** | 0.001 ** | 0.001 |
| | (−0.758) | (4.196) | (2.577) | (1.439) |
| $Yretwd_{i,t}$ | −0.004 *** | −0.004 *** | 0.002 *** | 0.004 *** |
| | (−5.526) | (−5.268) | (4.441) | (7.018) |
| $Rd_{i,t}$ | 0.139 *** | 0.068 ** | 0.011 | 0.010 |
| | (3.306) | (1.968) | (0.446) | (0.560) |
| $Dual_{i,t}$ | 0.008 | −0.007 ** | −0.005 | −0.001 |
| | (1.397) | (−2.542) | (−1.560) | (−0.669) |
| $Board_{i,t}$ | −0.000 | −0.000 ** | 0.000 | 0.000 |
| | (−0.891) | (−2.336) | (0.719) | (1.559) |
| $Indp_{i,t}$ | 0.001 | −0.000 | −0.004 | 0.002 |
| | (0.196) | (−0.104) | (−1.070) | (0.806) |
| $Constant_{i,t}$ | 0.148 *** | 0.057 *** | −0.056 *** | −0.058 *** |
| | (9.048) | (3.969) | (−5.538) | (−6.369) |
| Industry-fixed effects | Yes | Yes | Yes | Yes |
| Observations | 12,476 | 12,470 | 12,476 | 12,470 |
| Adjusted $R^2$ | 0.1087 | 0.1002 | 0.0367 | 0.0420 |

All models in Table 6 were estimated with industry-fixed effects. Reported in parentheses are the *t*-statistics based on robust standard errors clustered at the firm level. **, and *** indicate statistical significance at the 5%, and 1% levels, respectively.

## 5. Robustness Tests

### 5.1. Changes in the Measurement of Dependent Variables

To ensure robustness, we used the natural logarithm of the total investment to measure the investment level ($Loginv_{i,t+1}$). We also measured the investment efficiency ($Inveff1_{i,t+1}$) as the absolute value of the residual from the investment measurement model as in Biddle et al. (2009) [19], multiplied by −1.

Columns (1) and (2) of Table 7 present the empirical results after a change in the measurement of the dependent variables. As shown in columns (1) and (2), the coefficients of $Treat_{i,t}$ were 0.301 and −0.006, respectively, both significant at the 1% level. The coefficients of $Treat_{i,t} \times Event_{i,t}$ in columns (1) and (2) of Table 7 were −0.318 and 0.006, respectively, both significant at the 1% level. These robust results support Hypotheses 1 and 2.

**Table 7.** Alternative dependent variables and control year-fixed effects.

| | (1) | (2) | (3) | (4) |
|---|---|---|---|---|
| | $Loginv_{i,t+1}$ | $Inveff1_{i,t+1}$ | $Inv_{i,t+1}$ | $Inveff_{i,t+1}$ |
| $Treat_{i,t}$ | 0.301 *** | −0.006 *** | 0.015 *** | −0.003 * |
| | (5.841) | (−3.077) | (5.173) | (−1.838) |
| $Treat_{i,t} \times Event_{i,t}$ | −0.318 *** | 0.006 *** | −0.017 *** | 0.005 *** |
| | (−5.093) | (2.862) | (−5.732) | (2.895) |
| $Event_{i,t}$ | −0.377 *** | 0.006 *** | | |
| | (−13.881) | (9.029) | | |
| $Size_{i,t}$ | 0.988 *** | 0.001 | 0.001 | −0.001 ** |
| | (40.603) | (1.475) | (0.651) | (−2.548) |
| $Lev_{i,t}$ | −0.020 | −0.008 *** | 0.000 | −0.001 |
| | (−0.198) | (−4.147) | (0.064) | (−0.820) |
| $Roa_{i,t}$ | 0.880 *** | −0.009 *** | 0.026 *** | −0.007 *** |
| | (12.292) | (−6.218) | (11.896) | (−4.715) |
| $Mb_{i,t}$ | −0.021 *** | −0.001 *** | 0.000 | −0.001 *** |
| | (−3.905) | (−5.096) | (1.085) | (−5.756) |
| $Cash_{i,t}$ | 0.087 *** | 0.001 ** | 0.001 * | 0.001 *** |
| | (4.726) | (2.451) | (1.842) | (3.116) |
| $Yretwd_{i,t}$ | −0.047 *** | 0.003 *** | 0.000 | 0.001 * |
| | (−3.282) | (7.760) | (0.408) | (1.879) |
| $Rd_{i,t}$ | 5.408 *** | 0.010 | 0.051 * | 0.053 *** |
| | (5.711) | (0.599) | (1.766) | (3.189) |
| $Dual_{i,t}$ | 0.039 | −0.004 * | −0.000 | −0.002 |
| | (0.399) | (−1.650) | (−0.120) | (−0.997) |
| $Board_{i,t}$ | −0.004 | 0.000 | −0.000 ** | 0.000 ** |
| | (−0.844) | (1.344) | (−2.244) | (1.979) |
| $Indp_{i,t}$ | −0.028 | 0.000 | 0.004 | −0.003 |
| | (−0.251) | (0.150) | (0.983) | (−1.335) |
| $Constant_{i,t}$ | −4.513 *** | −0.074 *** | 0.040 *** | −0.035 *** |
| | (−13.226) | (−8.759) | (3.176) | (−4.813) |
| Industry-fixed effects | Yes | Yes | Yes | Yes |
| Year-fixed effect | No | No | Yes | Yes |
| Observations | 24,946 | 24,946 | 24,946 | 24,946 |
| Adjusted $R^2$ | 0.0617 | 0.0539 | 0.1148 | 0.0452 |

All models in Table 7 were estimated with industry-fixed effects. Reported in parentheses are the *t*-statistics based on robust standard errors clustered at the firm level. *, **, and *** indicate statistical significance at the 10%, 5%, and 1% levels, respectively.

### 5.2. Control Year-Fixed Effect

We controlled the industry-fixed effects in the main empirical test. This section controls both the industry-fixed effects and year-fixed effects. In Table 7, columns (3) and column (4) indicate that after controlling for the industry-fixed effect and year-fixed effect, the empirical results remained unchanged.

### 5.3. PSM Procedure

In order to alleviate the endogenous problem caused by selection deviation, we further undertook a propensity score matching (PSM) procedure to match pollutant-emitting firms to non-pollutant-emitting firms. Specifically, we used logit regression to estimate a firm's probability of emitting pollutants. We then matched the treatment group with the control group through the closest neighbor matching technique using ratios of 1:1 and 1:2. Following Cui et al. (2022) [8], we included eight control variables in the main regression: the firm assets (Size), the ratio of net income to total assets (Roa), the debt-to-asset proportion (Lev), the cash holdings (Cash), the book-to-market ratio (Mb), the research and development expenditure (Rd), the stock returns (Yretwd), the board size (Board), and control of the year-fixed effects and the industry-fixed effects. Table 8 indicates that after

the PSM procedure, the coefficients of the $Treat_{i,t}$ and $Treat_{i,t} \times Event_{i,t}$ were significant, and the results remained unchanged, suggesting that the main results are robust.

**Table 8.** PSM results.

| | (1) | (2) | (3) | (4) |
|---|---|---|---|---|
| | 1:1 | 1:1 | 1:2 | 1:2 |
| | $Inv_{i,t+1}$ | $Inveff_{i,t+1}$ | $Inv_{i,t+1}$ | $Inveff_{i,t+1}$ |
| $Treat_{i,t}$ | 0.017 *** | −0.004 ** | 0.016 *** | −0.004 ** |
| | (5.101) | (−2.302) | (5.063) | (−2.289) |
| $Treat_{i,t} \times Event_{i,t}$ | −0.016 *** | 0.007 *** | −0.016 *** | 0.006 *** |
| | (−3.870) | (2.972) | (−4.508) | (3.070) |
| $Event_{i,t}$ | −0.017 *** | 0.007 *** | −0.017 *** | 0.007 *** |
| | (−5.745) | (4.695) | (−7.722) | (5.161) |
| $Size_{i,t}$ | 0.000 | −0.002 ** | 0.000 | −0.002 ** |
| | (0.101) | (−2.097) | (0.016) | (−1.975) |
| $Lev_{i,t}$ | 0.006 | −0.002 | 0.001 | −0.001 |
| | (0.808) | (−0.433) | (0.076) | (−0.235) |
| $Roa_{i,t}$ | 0.049 *** | −0.014 *** | 0.041 *** | −0.011 *** |
| | (7.503) | (−3.691) | (7.261) | (−3.458) |
| $Mb_{i,t}$ | 0.001 | −0.001 *** | 0.001 | −0.001 ** |
| | (1.424) | (−2.685) | (1.102) | (−2.355) |
| $Cash_{i,t}$ | 0.001 | 0.002 * | 0.000 | 0.002 ** |
| | (0.437) | (1.886) | (0.117) | (2.398) |
| $Yretwd_{i,t}$ | −0.007 *** | 0.004 *** | −0.005 *** | 0.003 *** |
| | (−4.759) | (4.254) | (−4.427) | (3.671) |
| $Rd_{i,t}$ | 0.276 *** | −0.124 ** | 0.281 *** | −0.074 |
| | (2.676) | (−2.110) | (3.271) | (−1.475) |
| $Dual_{i,t}$ | 0.006 | −0.010 | 0.005 | −0.009 * |
| | (0.567) | (−1.455) | (0.592) | (−1.744) |
| $Board_{i,t}$ | 0.000 | 0.000 | −0.000 | −0.000 |
| | (0.004) | (0.074) | (−0.319) | (−0.264) |
| $Indp_{i,t}$ | −0.018 | 0.013 * | −0.014 | 0.009 * |
| | (−1.588) | (1.927) | (−1.485) | (1.737) |
| $Constant_{i,t}$ | 0.044 * | −0.011 | 0.060 *** | −0.022 |
| | (1.777) | (−0.659) | (2.735) | (−1.452) |
| Industry-fixed effects | Yes | Yes | Yes | Yes |
| Observations | 3232 | 3232 | 4848 | 4848 |
| Adjusted $R^2$ | 0.0879 | 0.0344 | 0.0753 | 0.0329 |

All models in Table 8 were estimated with industry-fixed effects. Reported in parentheses are the *t*-statistics based on robust standard errors clustered at the firm level. *, **, and *** indicate statistical significance at the 10%, 5%, and 1% levels, respectively.

### 5.4. The 2012–2020 Period

The "Green Credit Guideline" was issued by the China Banking Regulatory Commission in 2012 to encourage banks to allocate credit resources to green industry initiatives. To determine the effect of the 2012 Green Credit Guideline, we used the 2012 to 2020 period to reexamine our main results. In Table 9, columns (1) and (2) show that from 2012 to 2020, the coefficients of $Treat_{i,t} \times Event_{i,t}$ were significant at the 1% and 5% levels, respectively, which suggests that, excluding the effect of the 2012 Green Credit Guideline, the 2016 Green credit policy still significantly decreased the pollutant-emitting firms' investment level but improved their investment efficiency. The results in Table 9 suggest that in the 2012 to 2020 period, the results remained unchanged.

**Table 9.** The 2012–2020 period.

| | (1) | (2) |
|---|---|---|
| | 2012–2020 | |
| | Inv$_{i,t+1}$ | Inveff$_{i,t+1}$ |
| Treat$_{i,t}$ | 0.011 *** | −0.003 * |
| | (3.555) | (−1.794) |
| Treat$_{i,t}$ × Event$_{i,t}$ | −0.012 *** | 0.004 ** |
| | (−4.031) | (2.560) |
| Event$_{i,t}$ | −0.012 *** | 0.005 *** |
| | (−11.695) | (9.537) |
| Size$_{i,t}$ | −0.001 | −0.001 *** |
| | (−0.784) | (−2.888) |
| Lev$_{i,t}$ | 0.003 | 0.003 * |
| | (0.931) | (1.930) |
| Roa$_{i,t}$ | 0.025 *** | −0.006 *** |
| | (10.868) | (−4.259) |
| Mb$_{i,t}$ | 0.000 | −0.001 *** |
| | (0.552) | (−6.687) |
| Cash$_{i,t}$ | 0.001 ** | 0.001 *** |
| | (2.178) | (4.467) |
| Yretwd$_{i,t}$ | 0.000 | 0.002 *** |
| | (0.150) | (3.527) |
| Rd$_{i,t}$ | 0.084 *** | 0.006 |
| | (3.081) | (0.431) |
| Dual$_{i,t}$ | 0.001 | −0.003 ** |
| | (0.179) | (−1.985) |
| Board$_{i,t}$ | −0.000 *** | 0.000 |
| | (−3.063) | (0.621) |
| Indp$_{i,t}$ | 0.004 | −0.001 |
| | (1.045) | (−0.338) |
| Constant$_{i,t}$ | 0.056 *** | −0.032 *** |
| | (4.311) | (−6.457) |
| Industry-fixed effects | Yes | Yes |
| Observations | 19,450 | 19,450 |
| Adjusted $R^2$ | 0.0986 | 0.0369 |

All models in Table 9 were estimated with industry-fixed effects. Reported in parentheses are the *t*-statistics based on robust standard errors clustered at the firm level. *, **, and *** indicate statistical significance at the 10%, 5%, and 1% levels, respectively.

## 6. Conclusions

Currently, the public and scholars are paying increasing attention to ESG practices. As an important part of ESG practice, the green credit policy aims to reduce pollution by allocating more credit to green industries. Since 2007, the Chinese government has promulgated several policies to encourage commercial banks to issue green credits. The Chinese 2016 Green credit policy made it a compulsory requirement for commercial banks to limit pollutant-emitting firms' credits. Previous studies suggested that firms without sufficient bank loans would decrease their investment [16,17], but it was unknown whether they would make better use of the limited bank loans to improve their performance. This paper used Chinese A-share-listed firms to investigate whether the 2016 Green credit policy affected pollutant-emitting firms' investment decisions. This paper hypothesized that after the 2016 Green credit policy, firms with limited credits would reduce their investment expenditure and would be cautious regarding their investment, as they would not enjoy sufficient credit from commercial banks; thus, firms would use their limited credit more effectively, which could improve their investment efficiency. Consistent with this hypothesis, the 2016 Green credit policy can reduce pollutant-emitting firms' investment level but significantly improve their investment efficiency. These findings suggest that the Chinese Green credit policy affected firms' behaviors positively, which is different

from the conclusion by Wang et al. (2020) [11] and Hao et al. (2020) [43], who found that the green credit policy increased firms' financial pressure and decreased their investment level, leading to a negative effect on firms' behaviors. Further analysis showed that the 2016 Green credit policy had a more pronounced effect on state-owned firms, firms with high-quality corporate governance, and those with a high level of analyst following.

This paper has several policy implications. As a compulsory policy, Chinese green credit policy can improve pollutant-emitting firms' investment efficiency, but this positive effect is heterogeneous. Other countries may consider changing their voluntary green credit policy to a compulsory policy. For example, developing countries can require banking regulators to make a mandatory evaluation of commercial banks' green credit allocation, which can make the process more effective. Commercial banks and other financial institutions should consider different characteristics of pollutant-emitting firms, such as property rights, corporate governance, and analyst following, when they provide green credits. For example, when commercial banks issue green credits to firms, they should not only consider firms' intended use of the credits but also the firms' corporate governance and other characteristics.

This paper fills the literature gap left by existing studies. However, this paper also has several limitations, which can be supplemented by future research. This paper showed that the Chinese green credit policy can affect pollutant-emitting firms' investment level and investment efficiency. However, it is still unknown whether and how firms use the credit to change their investment portfolio after the green credit policy. Some studies have analyzed the performance of ESG-investing portfolios [44–46], but few have investigated whether and how firms would use green credits to invest in an ESG portfolio; future research can fill this gap. In addition, this paper found that the compulsory green credit policy in China affected firms' investment decisions; the impact of a compulsory green credit policy on firms' investment decisions will be different in different countries, as future research can explore.

**Author Contributions:** Conceptualization, X.L., D.D. and L.Y.; methodology, X.L. and L.Y.; formal analysis, X.L. and L.Y.; investigation, X.L., D.D. and L.Y.; resources, X.L., D.D. and L.Y.; data curation, X.L. and L.Y.; writing—original draft preparation, X.L.; writing—review and editing, X.L. and L.Y.; visualization, X.L. and L.Y.; supervision, D.D.; project administration, L.Y. and D.D.; funding acquisition, L.Y. All authors have read and agreed to the published version of the manuscript.

**Funding:** This research was funded by the Youth Project of Beijing Social Science Foundation (grant number: 21GLC037).

**Institutional Review Board Statement:** Not applicable.

**Informed Consent Statement:** Not applicable.

**Data Availability Statement:** The datasets used and/or analyzed during the current study are available from the corresponding author upon reasonable request.

**Acknowledgments:** We are grateful to the funding agencies.

**Conflicts of Interest:** The authors declare no conflict of interest.

# Appendix A

**Table A1.** Pearson's correlation coefficient and Spearman's rank correlation.

| | (1) | (2) | (3) | (4) | (5) | (6) | (7) | (8) | (9) | (10) | (11) | (12) |
|---|---|---|---|---|---|---|---|---|---|---|---|---|
| (1) $Inv_{i,t+1}$ | 1 | −0.208 *** | 0.012 ** | −0.076 *** | 0.128 *** | 0.002 | 0.021 *** | 0.018 *** | −0.014 ** | −0.003 | −0.002 | 0.033 *** |
| (2) $Inveff_{i,t+1}$ | −0.658 *** | 1 | 0.028 *** | 0.004 | −0.019 *** | −0.012 ** | 0.046 *** | 0.028 *** | 0.031 *** | −0.010 | 0.015 ** | −0.030 *** |
| (3) $Treat_{i,t}$ | −0.004 | 0.043 *** | 1 | 0.449 *** | 0.180 *** | −0.500 *** | 0.769 *** | −0.048 *** | 0.049 *** | 0.072 *** | 0.216 *** | −0.073 *** |
| (4) $Event_{i,t}$ | −0.039 *** | −0.003 | 0.444 *** | 1 | −0.053 *** | −0.137 *** | 0.202 *** | −0.037 *** | −0.090 *** | 0.032 *** | 0.121 *** | −0.026 *** |
| (5) $Size_{i,t}$ | 0.100 *** | −0.024 *** | 0.125 *** | −0.186 *** | 1 | 0.116 *** | 0.231 *** | 0.135 *** | −0.011 * | −0.003 | −0.021 *** | 0.031 *** |
| (6) $Lev_{i,t}$ | −0.030 *** | −0.047 *** | −0.404 *** | 0.026 *** | −0.154 *** | 1 | −0.401 *** | 0.382 *** | −0.176 *** | −0.027 *** | −0.103 *** | 0.110 *** |
| (7) $Roa_{i,t}$ | 0.002 | 0.055 *** | 0.805 *** | 0.203 *** | 0.172 *** | −0.362 *** | 1 | −0.053 *** | 0.039 *** | 0.067 *** | 0.157 *** | −0.042 *** |
| (8) $Mb_{i,t}$ | 0.002 | 0.007 | −0.080 *** | −0.024 *** | 0.100 *** | 0.353 *** | −0.091 *** | 1 | −0.090 *** | 0.026 *** | 0.033 *** | 0.166 *** |
| (9) $Cash_{i,t}$ | −0.025 *** | 0.041 *** | −0.065 *** | −0.118 *** | −0.051 *** | −0.026 *** | −0.028 *** | −0.030 *** | 1 | 0.099 *** | 0.062 *** | −0.209 *** |
| (10) $Yretwd_{i,t}$ | −0.005 | −0.003 | 0.081 *** | 0.032 *** | −0.019 *** | −0.014 ** | 0.075 *** | 0.013 ** | 0.075 *** | 1 | 0.053 *** | −0.025 *** |
| (11) $Rd_{i,t}$ | −0.015 ** | 0.018 *** | 0.226 *** | 0.124 *** | −0.050 *** | −0.059 *** | 0.167 *** | 0.021 *** | 0.020 *** | 0.056 *** | 1 | −0.167 *** |
| (12) $Dual_{i,t}$ | 0.033 *** | −0.026 *** | −0.052 *** | −0.013 ** | 0.051 *** | 0.083 *** | −0.034 *** | 0.147 *** | −0.138 *** | −0.017 *** | −0.076 *** | 1 |

This table shows the Spearman correlation coefficients in the upper right and the Pearson correlation coefficients in the lower left. All the correlation tests were two-tailed. *, **, and *** indicate statistical significance at the 10%, 5%, and 1% levels, respectively. There was no evidence of multicollinearity if the VIF value as below 5, the critical level, according to Gujarati (2003) [37].

# Appendix B

**Table A2.** Variance inflation factors.

| | Equation (1) | | Equation (2) | |
|---|---|---|---|---|
| | VIF | 1/VIF | VIF | 1/VIF |
| Treat | 1.79 | 0.56 | 1.79 | 0.56 |
| Treat × Event | 1.68 | 0.59 | 1.68 | 0.59 |
| Event | 1.68 | 0.60 | 1.68 | 0.60 |
| Size | 4.86 | 0.21 | 4.86 | 0.21 |
| Lev | 1.83 | 0.55 | 1.83 | 0.55 |
| Roa | 1.16 | 0.86 | 1.16 | 0.86 |
| Mb | 1.55 | 0.64 | 1.55 | 0.64 |
| Cash | 3.32 | 0.30 | 3.32 | 0.30 |
| Yretwd | 1.33 | 0.75 | 1.33 | 0.75 |
| Rd | 1.26 | 0.79 | 1.26 | 0.79 |
| Dual | 1.02 | 0.98 | 1.02 | 0.98 |
| Board | 1.09 | 0.91 | 1.09 | 0.91 |
| Indp | 1.14 | 0.87 | 1.14 | 0.87 |
| $\sum$Industry | 3.08 | 0.51 | 3.11 | 0.52 |
| Mean VIF | 1.91 | | 1.92 | |

## Appendix C

**Table A3.** A detailed composition of the corporate governance (CG) level according to Wu et al. (2020) [41].

| Variable | Definition |
| --- | --- |
| CG | Corporate governance level is the sum of 14 internal governance factors. |
| (1) Strategy Committee Size | An indicator variable that equals 1 if the size of an enterprise's strategy committee is larger than the sample median, and 0 otherwise. |
| (2) Nomination Committee Size | An indicator variable that equals 1 if the size of an enterprise's nomination committee is larger than the sample median, and 0 otherwise. |
| (3) Audit Committee Size | An indicator variable that equals 1 if the size of an enterprise's audit committee is larger than the sample median, and 0 otherwise. |
| (4) Compensation Committee Size | An indicator variable that equals 1 if the size of an enterprise's compensation committee is larger than the sample median, and 0 otherwise. |
| (5) Independence of Strategy Committee | An indicator variable that equals 1 if the ratio of an enterprise's independent directors on the strategy committee is larger than the sample median, and 0 otherwise. |
| (6) Independence of Nomination Committee | An indicator variable that equals 1 if the ratio of an enterprise's independent directors on the nomination committee is larger than the sample median, and 0 otherwise. |
| (7) Independence of Audit Committee | An indicator variable that equals 1 if the ratio of an enterprise's independent directors on the audit committee is larger than the sample median, and 0 otherwise. |
| (8) Independence of Compensation Committee | An indicator variable that equals 1 if the ratio of an enterprise's independent directors on the compensation committee is larger than the sample median, and 0 otherwise. |
| (9) Number of Supervisory Board members | An indicator variable that equals 1 if the number of an enterprise's supervisory board members is larger than the sample median, and 0 otherwise. |
| (10) Expertise of Supervisory Board | An indicator variable that equals 1 if the number of an enterprise's committee members with finance and accounting expertise is larger than the sample median, and 0 otherwise. |
| (11) Share ownership of Supervisory Board members | An indicator variable that equals 1 if the share ownership of an enterprise's supervisory board members is larger than the sample median, and 0 otherwise. |
| (12) Number of Supervisory Board meetings | An indicator variable that equals 1 if the number of an enterprise's supervisory board meetings is larger than the sample median, and 0 otherwise. |
| (13) Share ownership of the largest and second-largest shareholders | An indicator variable that equals 1 if the share ownership of an enterprise's largest shareholder and the second-largest shareholder is larger than the sample median, and 0 otherwise. |
| (14) Share ownership of the largest shareholder | An indicator variable that equals 1 if the share ownership of an enterprise's largest shareholder is larger than the sample median, and 0 otherwise. |

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
