# Peer review of "Green Credit Policy and Investment Decisions: Evidence from China"

_sustainability, doi:10.3390/su14127088_

Round 1

Reviewer 1 Report

I like this work. I like the discussion. It is a good job but needs a little bit of shape in the results discussion and its relation with the current knowledge.

Your paper is relevant and very focused on the Chinese case. That is alright but your paper has a lot of potential to be discussed in the Global Academia, highlighting the results in China with others in other countries and specific industries.

Therefore I will give a detailed and mandatory list of suggestions to enhance your paper. That is my intention with these to reach not a good but a great paper:

  1. In line 64 you present the PSM procedure but you do not define it (it is the first time it is quoted) for the unrelated reader. Please, correct.
  2. The same issue happens in line 221. Please, define DID for the unrelated reader.
  3. You didn't define what the variables Industry_i are. Are these dummies of industry-specific control variables? this happens in Table 1, (1), and (2). Please, correct and give a strong foundation about their use.
  4. Why the leverage effect (event) is not used in the other variables in (1) and (2). Your degrees of freedom seem to be sufficient. Please explain and give a foundation for this leverage effect (X_i * Event) in the corresponding factor variables.
  5. For multicollinearity, you must use something more robust than the mere Pearson or Spearman correlations. Thoe are two-way concordance metrics and say nothing about the general incidence of a factor with the other ones in general multivariate correlation terms. Variance inflation factors are a good answer that you must use in your paper, complementary to the corresponding correlation tables in your appendix. Please, enhance.
  6. In Figure 1, please change the units of the Y-axis to % if these are either logarithms or proportions against equity, cash flow, sales, or something else.
  7. You don't discuss figures 1 and 2. What should the reader infer and conclude from these?
  8. In Table 4 you are presenting the t-statistic (please, correct the capital "T") in parentheses. I want to suggest of omit the use of parentheses or please the coefficient's standard error instead.
  9. Please explain how did you process the corporate governance level index (formulas and details briefly) for the unrelated reader. Is it a personal calculation method or have you seen that with other authors or methodologies. Please expand your method and estimation method fundaments.
  10. In Tables 4 and 5, you split your sample and analysis with either Soes_{i,t} or Cig_{i,t}. Why you didn't use a mere dummy variable? If you split the sample with these will the samples have the same size (that is degrees of freedom)? Wouldn't be useful to use dummy control variables of those factors (Soes_{i,t} or Cig_{i,t}) to avoid some other issues such as higher or lower heteroskedasticity or multicollinearity reduction? Please, expand or correct if necessary. Note: the same happens with Table 6.
  11. The English writing style is quite good in the paper. Still, I suggest making an additional review. The paragraph in lines 367 to 380 is a little bit long and wordy. It is slow (and a little bit hard) to follow. I believe that his paragraph is a very important one in the results review. please adapt the paragraph.
  12. You have made a very interesting and strong review of your results. I believe that it will be necessary two things in your results review:
    1. Please give the analysis route of all the tests that you present in the results review section and give a brief foundation of why are you using those tests. I suggest making this brief route compared with the previous literature, stating why are you doing that and what you want to improve or extend in the results of those previous works.
    2. Before the conclusions section and given the wide results review, please make a corollary of your results in a corollary of results sob-section. This will be a preamble for your conclusions, it will help to highlight your findings and will help the reader to have a detailed but summarized view of your results and findings.
  13. Please, your literature review section is quite good but it must be discussed in this way in each reference or group of references:
    1. What has been done in that or those works?
    2. How has been done?
    3. What are the main results and findings?
    4. What could be the drawbacks or issues that you want to extend or improve in your work? That is, how does this work or works motivate your work.
  14. Following the previous suggestion, your results must be discussed in comparison with the previous works that your quote in the literature review section. With this, you will enhance the Academic discussion that is intended in your work and is the journal's main goal. please adapt your paper.
  15. Your literature review is quite good but you must widen the implications of your important results. More specifically, in ESG or socially responsible investing. If Chinese companies have an impact on their access to credit due to environmental issues and policies, this will have an impact on company (and stock price) performance. Therefore, you have a subtle but important link between ESG investment in your conclusions and the impact of your work. Please, enhance a little your literature review by linking your work with this important subject of ESG investing. please, keep in mind that the corporate discussion is (Milton Friedman) that being environmentally friendly has a negative impact on profits (one side), and the profits and company performance benefit is in the long term and these companies are more stable in their profits, less risky (Freeman). I suggest you some references that must be in your paper as a departing set of references:
    1. Banking credit and ESG credits: https://doi.org/10.1016/j.mulfin.2019.01.002
    2. Sovereing debt and ESG: http://link.springer.com/10.1007/s10551-010-0638-3
    3. The impact of good ESG practices on bank stock prices: https://onlinelibrary.wiley.com/doi/10.1002/csr.1759
    4. A not so strong relation between being an environmentally responsible and the impact on the environment in Europe (a contrary position to your paper): https://www.mdpi.com/2071-1050/12/23/9855
    5. The benefits for institutional investors (pension funds) of allocating to socially responsible (environmentally friendly) stocks: https://doi.org/10.3390/su11010178
  16. Please enhance your conclusions:
    1. Is your most important section. please discuss briefly what have you done, what are your main findings,
    2. How do you contribute to the current knowledge on the subject (in China and, most of all, abroad)?
    3. What are the similarities (coincidences)  and new findings, compared with previous works? 
    4. What are the limitations of your works and the guidelines for further research that you suggest?

I hope that these suggestions will be of use to enhance your good work.

Reviewer 2 Report

There are only two hypotheses  so there should be more. Paragraphs are so lengthy they should be in the limit. 

In Table 2:Descriptive statistics of all the variables is given but only mean of two variables is discussed so please give details of all the descriptive for all the variables. 

In Table 3: Adj R^2 for two models is very low plz rectify it and cover it in detail of the existing literature. Same case in Table 4 and 5 respectively. 

Reviewer 3 Report

your paper is excellent, I rarely give this feedback. Good job.

Round 2

Reviewer 1 Report

The paper is a complete and exciting review of the effects of compulsory legal requirements to finance ESG companies in China. Despite the authors' full review, tests, and conclusions about the topic, I believe that the authors can and must expand their previous and motivating literature review.

ESG laws and their impact on company performance is a topic widely discussed. The effect of funds and Banks that promote ESG investing in companies with sound environmental practices is a topic that should be briefly discussed in the paper. The author must link this discussion with the paper’s motivations.

Also, the discussion of the ESG quality of a given company with its risk (total or systematic) is a type of literature related to the paper.

Therefore, the authors must expand the literature about banking ESG credit regulation and the relation of ESG quality with profitability, company performance, and risk level. It is essential to give a wither and the punctual context of how the paper contributes to knowledge.

Also, the paper needs some improvements in English writing style.

Author Response

Reviewer’s Comments to Author

Response

Thank you for the comment.

We have expanded the literature about banking ESG credit regulation and the relation of ESG quality with company performance, and risk level. Specifically,on the introduction, we expand the literature about the relation of ESG quality with company performance and risk level. And we add the “2.2 literature review”. In the literature review, we have expanded the literature about the banking ESG credit regulation.

Also, we use the editing service to make the English editing revisions. We hope that, after revised by the professional editing service, the English writing style of our manuscript has been improved.

Please also kindly see the attachment.
